# Experimental Analysis of the Annular Velocity of a Capsule When Starting at Different Positions of a Horizontal Bend Pipe

**Cheng Wang [1,2] and Xihuan Sun [1,*]**

[1] College of Water Resources Science and Engineering, Taiyuan University of Technology, Taiyuan 030024, China
[2] Polytechnic Institute, Taiyuan University of Technology, Xiaoyi 032300, China
* Correspondence: sunxihuan@tyut.edu.cn; Tel.: +86-13513600012

**Abstract:** The study of the annular slit flow field is important for energy consumption, transport efficiency, and the force on the capsule for hydraulic capsule transportation. A combination of physical experiments and theoretical analysis was used to study the annular flow field around a capsule that was set in motion at different positions of a horizontal bend pipe. We study the flow velocity distribution of the gap flow field at different bend positions of the capsule by changing the position of the capsule at the bend. We found that the distribution of the flow field remained similar for different starting positions of the capsule, but the flow velocity increased suddenly and dramatically at the inflow section of the ring gap. We recorded different velocity distributions of the annular gap on the concave and convex sides of the pipe; on the convex side, the streamline of the gap was smooth, and the change in velocity was relatively small. The flow velocity of the slit flow varied more notably on the concave side of the pipe, and there was a greater fluctuation in the flow velocity distribution. Because the effects of the capsule and the pipe on water flow were not the same, we found large fluctuations in gap flow velocity at different measuring points on the concave side. Gap flow velocity was most influenced by axial flow velocity. We found that the axial flow velocity was about one order of magnitude greater than the radial flow velocity or circumferential flow velocity. In this paper, we analyze the changes in the ring gap flow field of the capsule at different bending positions and analyze the reasons for the flow field changes and the flow velocity distribution law. This is of great significance to the study of the transport efficiency and energy consumption of the capsule. The results of this paper complement the study of capsule initiation at different positions in the bend and provide a reference point in terms of transport efficiency, energy consumption, and capsule stress. The results of this study promote the development of hydraulic capsule transportation.

**Keywords:** hydraulic transportation; capsule; flow velocity; physics experiments

## 1. Introduction

As the logistics industry continues to develop in contemporary society, the disadvantages of traditional methods of transportation by road, rail, air, and water, including high pollution and high energy consumption, are becoming more and more obvious. The search for a new type of environmentally protective and energy-saving transportation mode is now an urgent matter. To this end, Professor Xihuan Sun [1] of Taiyuan University of Technology proposed a new type of hydraulic capsule pipeline transportation. This kind of technology offers the prospect of a mass transit method with low energy consumption and low pollution, as well as other advantages such as a small carbon footprint and minimal weather interference. To date, research on hydraulic capsule transportation in China and elsewhere has covered a number of study areas, including the influence of capsule structure on the flow velocity field during the hydraulic transportation process [2,3]. For example, the diameter ratio, volume ratio, and diameter–length ratio of the capsule will have different effects on the flow velocity distribution in the annular gap flow field and

the influence of hydraulic factors in the flow field on the operation of capsules [4,5], such as the Reynolds number, will affect the flow state of water flow in the gap flow field. Numerical simulation techniques have been used to analyze the flow field around capsules in pipelines [6]. For example, the FLUENT software is used to simulate the flow velocity field near the capsule, etc. The study of the flow field of the capsule annular gap flow is an important branch of hydraulic capsule pipeline transportation. The operating state of the capsule and the dual action of the pipe wall have different effects on the water flow field, and the characteristics of the water flow field are the main aspects that determine the transport efficiency of hydraulic capsule pipeline transportation. The water flow field at the entrance and exit of the slit is strongly disturbed, and the head loss along and locally forms an important part of the energy loss in the capsule transport process. The distribution of the annular gap flow field, especially the velocity distribution of the annular gap flow near the capsule wall, determines the magnitude of the viscous shear force on the wall to which the capsule is subjected. Therefore, it is important to increase the research on the capsule annular gap flow field for capsule transportation efficiency, energy consumption, and force. Xianliang Xu [7,8] used the Navier–Stokes equations (N–S equations) to derive differential equations for gap flow in spherical and flat surfaces without accounting for mass and inertial forces, with the viscous forces as the dominant factor, and further derived the equations for the pressure distribution in the gap flow. Unfortunately, Xu only theoretically derived the gap flow field mathematically and did not perform numerical simulations or physical experiments to verify the results, and its correctness has been questioned by some scholars. Wu et al. [9] used numerical simulations to study the internal field characteristics of annular slit injectors by controlling the structural parameters of the injectors and thus obtained the distribution characteristics of the annular gap flow field under different structural parameters of the injectors. Kim et al. [10] studied the flow characteristics of the runner gap in a mixed-flow turbine model and derived the relationship between gap flow velocity and gap width. Yang et al. [11] studied wall shear stresses in static-boundary annular slit flow and determined the distribution of these stresses using a model test method. Xiaomeng Jia [12] studied the annular gap flow field of a static-boundary capsule using the FLUENT software, and systematically analyzed the pressure distribution, flow velocity distribution, and vortex characteristics of the annular gap flow field. By such means, laws of pressure, flow velocity, and vortex characteristics relating to the gap flow field were obtained. Ihab et al. [13] used numerical simulation to study the pressure field inside the annular gap between the capsule and the pipeline and derived a distribution law for pressure within the annular gap flow. Pual et al. [14] performed computational fluid dynamics (CFD) modeling of a three-dimensional concentric capsule train to study the effects of capsule pitch ratio, diameter ratio, and aspect ratio on the energy consumption of capsule transportation, and the optimal capsule type was derived by optimal cost analysis. Lu [15] et al. used numerical simulation and theoretical analysis to study the threshold of the motion process of a wheeled capsule in the bend and concluded that the magnitude of the water flow velocity in the internal and external closed parts of the wheeled capsule is asymmetric, which provides implications for the study of energy consumption of wheeled capsule. These scholars started to build physical simulations and develop simulations of the slit flow field with various CFD software. However, CFD software itself is not exactly the same as the actual shortcomings, and the results of the study still have a small deviation from the actual physics. Li et al. [16] analyzed the forces acting upon the capsule and obtained a three-dimensional flow velocity distribution law for the dynamic-boundary annular slit flow. Jia et al. [17] constructed a physical test model to determine the distribution of annular slit flow boundary stresses and their laws under different Reynolds number conditions. Asim et al. [18] studied the effects of different shapes of capsules on the gap flow field and found that a lower shape factor led to a more uniform gap flow field, a lower drop in pressure, and reduced energy consumption. The bend section is an essential part of pipeline composition in pipeline transportation. Most of the studies by many scholars on the transport of capsules in pipelines are focused on straight pipe sections, and there

are fewer studies on the flow field distribution near the capsules in the bent pipe sections. Because physical experiments can demonstrate a more accurate actual flow field, more and more scholars are using physical experimental simulations to study the annular slit flow field. With the deepening of experimental research on physical models, many scholars have found that the flow in curved pipe sections is more complicated than that in straight pipe sections. However, limited by the complex flow conditions of curved pipes, few scholars have studied the flow velocity distribution in the slit flow field of a curved pipe section.

Previous research on crevice flow is also of value for the purposes of this study. Liu et al. [19] analyzed the basic hydraulic behavior of coal source flows in pipeline bends and found that hydraulic characteristics in bends were much more complex compared with straight sections of the pipeline. However, there have been few studies on capsule slit flow in bend sections to date. The more complex flow field distribution and the higher energy loss [20,21] in bend sections are matters which must be considered in the development of a slit flow theory of capsules. In the study reported here, we combined physical experiments with theoretical analysis to analyze the flow velocity distribution of the gap flow field when capsules were set in motion at different positions of a horizontal bend. Compared with previous studies, this paper focuses more on the actual flow field as demonstrated by the physical model. Although previous researchers have done physical experiments on the gap flow field of the capsule at the bend, the capsule selected in this study is different from the capsules in the previous studies. The capsule in this study has a unique foot design, which can form a stable concentric annular gap flow field and solve the problem of wear and collision and high resistance for the inner wall of the pipe during capsule transportation. Our results provide a firm theoretical foundation for future engineering applications of hydraulic capsule transportation technology.

## 2. Experimental System and Test Protocol

### 2.1. Experimental System Arrangement

The experimental system was composed of three parts: a power and regulation system, a test piping system, and a test system. The power and regulation system consisted of centrifugal pumps, gate valves, and electromagnetic flow meters. The test piping system comprised stainless steel round tubes and transparent Plexiglas round tubes. The pipeline as a whole was divided into horizontal straight sections and horizontal bent sections. The test system mainly involved the use of particle image velocimetry (PIV). Figure 1 shows specific components of the experimental system arrangement.

Before testing, the steel tank was filled with water, an appropriate quantity of tracer particles was added, and the centrifugal pump was started to pump the water from the steel tank into the pipe. The flow rate in the pipe was adjusted using the gate valve and observed using the electromagnetic flow meter. This rate was set to a low level (20 m$^3$/h), and the capsule was placed into the pipeline system through the drop device. The drop device was then closed, the flow rate was increased appropriately, and the start device was opened. The capsule was moved to the measured section and secured. The valve was then adjusted until the required starting flow rate in the pipeline was attained. The water flow in the pipeline was then stabilized. We use the double-pulse laser in the PIV system to irradiate the capsule with laser along the horizontal lateral center profile at different positions of the bent tube. The good reflectivity of the tracer particles is used to reflect the laser light to the camera lens, and the camera continuously exposes to take pictures while using photographs to record the distance the particles move during the exposure time. The velocity of particle motion can be obtained by the equation $V = \frac{\Delta x}{\Delta t}$ (v is the velocity of particle motion; $\Delta x$ is the distance travelled by the particle during the exposure time; and $\Delta t$ is the camera exposure time). We can obtain the distribution of the flow field by the trajectory and velocity of the particle motion. After that, the flow rate was increased again, and the capsule was released into the recovery device, thus completing the whole test process. The water then flowed back into the tank, thus completing a circular closed loop.

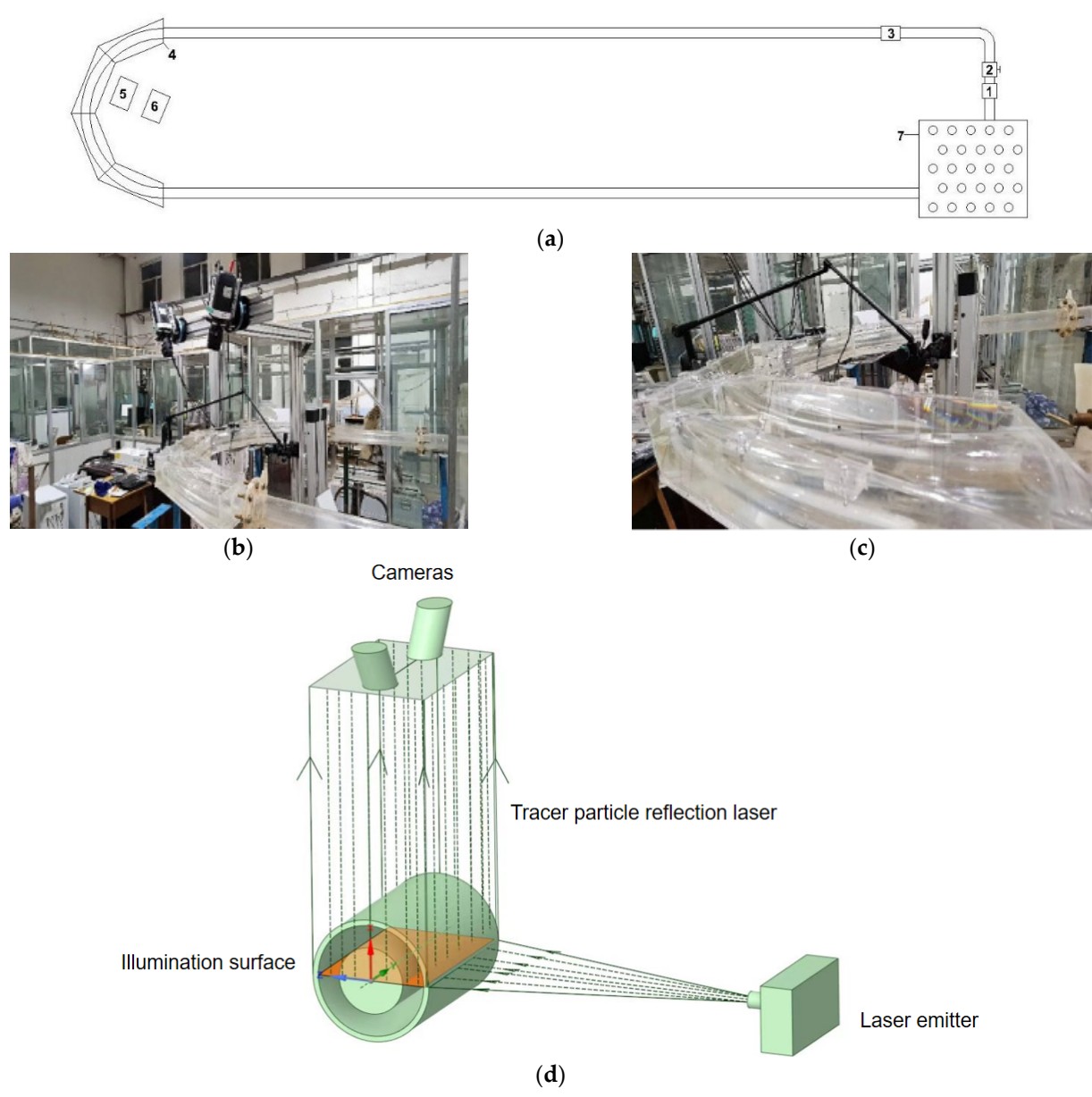

**Figure 1.** Layout of the test system: (**a**) schematic diagram, 1. centrifugal pumps, 2. gate valve, 3. electromagnetic flowmeter, 4. trapezoidal water jacket, 5. PIV, 6. workbench, 7. steel water tank; (**b**) PIV measuring; (**c**) bend pipe; (**d**) PIV schematic sketch.

The settings of the PIV system are shown in Table 1. Additionally, the processing of the PIV data was the same as that of Zhao et al. [22].

**Table 1.** Experimental settings for PIV measurements.

| Experimental Setting | Main Parameter |
| --- | --- |
| Illumination | Dual Power Nd-YLF Laser (2 × 30 mJ) |
| Camera lens | 2 Imager pro HS cameras |
| Image dimension | 2016 × 2016 pixels |
| Interrogation area | 32 × 32 pixels |
| Time between pulses | $5 \times 10^3$ µs |
| Seeding material | Polystyrene particles diameter 55 µm |
| Resolution ratio | 39.68 µm/pixel |

## 2.2. Capsule Structure

The capsule was 50 mm × 150 mm (diameter × length) in size and consisted of a material cylinder with support bodies and universal balls. The material cylinder formed the main material-loading component of the whole capsule body and was sealed with caps at both ends so that a closed space was formed. Support bodies were installed at 120° intervals at both ends of the material cylinder, making 6 support bodies in total. The role of the support bodies was to make the center axis of the capsule coincide with the center axis of the pipe so that the capsule could run smoothly in the pipe and produce a stable annular gap flow. Finally, universal balls were installed at the ends of the support bodies to reduce the frictional resistance between the capsule and the pipe wall. The structure diagram of the capsule is shown in Figure 2.

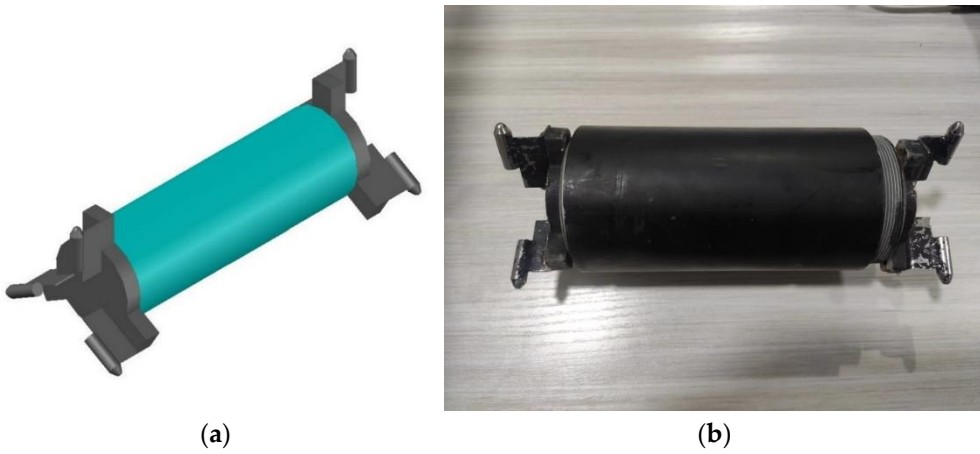

(**a**)                      (**b**)

**Figure 2.** Structure schematic of the capsule. (**a**) Capsule geometry model; (**b**) Capsule physical model.

## 2.3. Experimental Protocol Design

For the study of hydraulic capsule transportation technology, the previous research had divided the capsule state into three categories: static state, critical start state, and motion state [23]. The study of the critical start state determines the successful start of the capsule and is crucial to its proper operation. The principal purpose of the test was to assess the hydraulic characteristics of the loop gap water flow when the capsule was set in motion at different positions of the horizontal bend pipe. We termed the moment when the capsule was capable of movement but not yet in motion as the start of the capsule; the flow of water at the corresponding moment was termed the starting flow of the capsule. The Reynolds number is related to the water discharge $Q$ in the pipe, the diameter of the pipe $D$, and the kinematic viscosity $v$, and it can be expressed as $R_e = \frac{4Q}{\pi D v}$. We replaced the starting flow of the capsule with the Reynolds number to eliminate the influence of dimension.

The starting flow of the capsule in the bend was measured by placing the capsule inside the bend and changing the flow rate in the pipe by controlling the gate valve. In order to measure the flow velocity distribution when starting at different positions of the bend, we used the end of the upstream straight pipe section as the 0° position of the bend section. The capsule was then rotated counterclockwise and placed at four positions below the bend, as follows: (1) position 22.5°; (2) position 67.5°; (3) position 112.5°; (4) position 157.5°. The location at the bend is selected in a variety of ways in order to observe, as far as possible, the various slit flow fields that may arise at the bend. We split the 180° bend. Since the inlet and outlet of the bend are connected to a horizontal straight section, the error is extremely large for studying the slit flow in the bend. Therefore, we selected a location 22.5° from the inlet and outlet and the rest of the locations at 45° intervals. This avoids the influence of straight pipe sections at the inlet and outlet while selecting as many locations as possible for the bends and generalizing the majority of conditions. Figure 3

illustrates these specific positions schematically. As stated above, the capsule size was 50 mm × 150 mm (diameter × length), and the load was 1400 g. The pipe diameter was 100 mm.

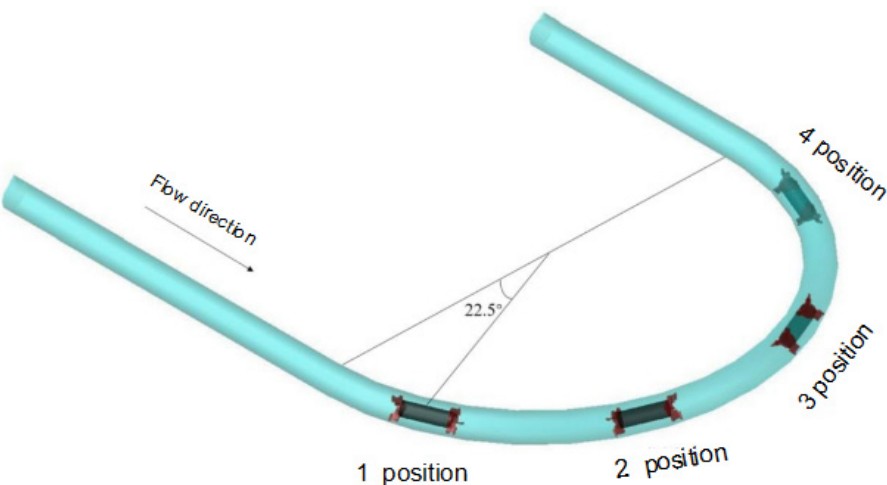

**Figure 3.** Schematic diagram of capsule at different positions of elbow.

### 2.4. Measurement Point Arrangement

When the capsule was in the bend position, the capsule axis did not coincide with the center of the bend cross-section and was an eccentric ring. In order to better study the variation in the slit flow field in the vicinity of the capsule, the circular slit flow measurement points were arranged in such a way that the center of the capsule axis was the center of the ring. For this test, annular gap flow measurement points were set with the capsule axis as the center of the ring circle and ring radii of 30 mm, 35 mm, 40 mm, 42 mm, and 45 mm, making a total of 5 rings. Measurement points were arranged using polar coordinates, with polar axes set at 30° intervals, making a total of 12 axes. The intersection of the measurement ring and the polar axis was used as the measurement point, and the ring slit was arranged with a total of 60 measurement points, as shown in Figure 4.

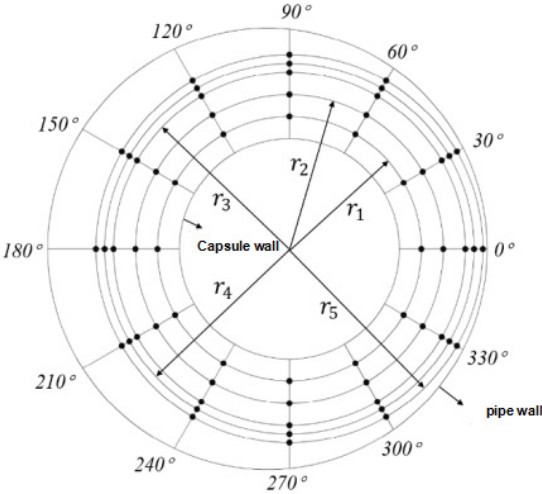

**Figure 4.** Schematic diagram of measuring point layout.

## 3. Results and Analysis

### 3.1. Circumferential Flow Velocity Characteristics

Using our 1400 g loaded capsule, we analyzed the overall distribution of the annular gap flow velocity when the capsule started at different positions of the bend (22.5°, 67.5°,

112.5°, 167.5°). A cloud chart of the time-averaged distribution of overall capsule velocity is shown in Figure 5.

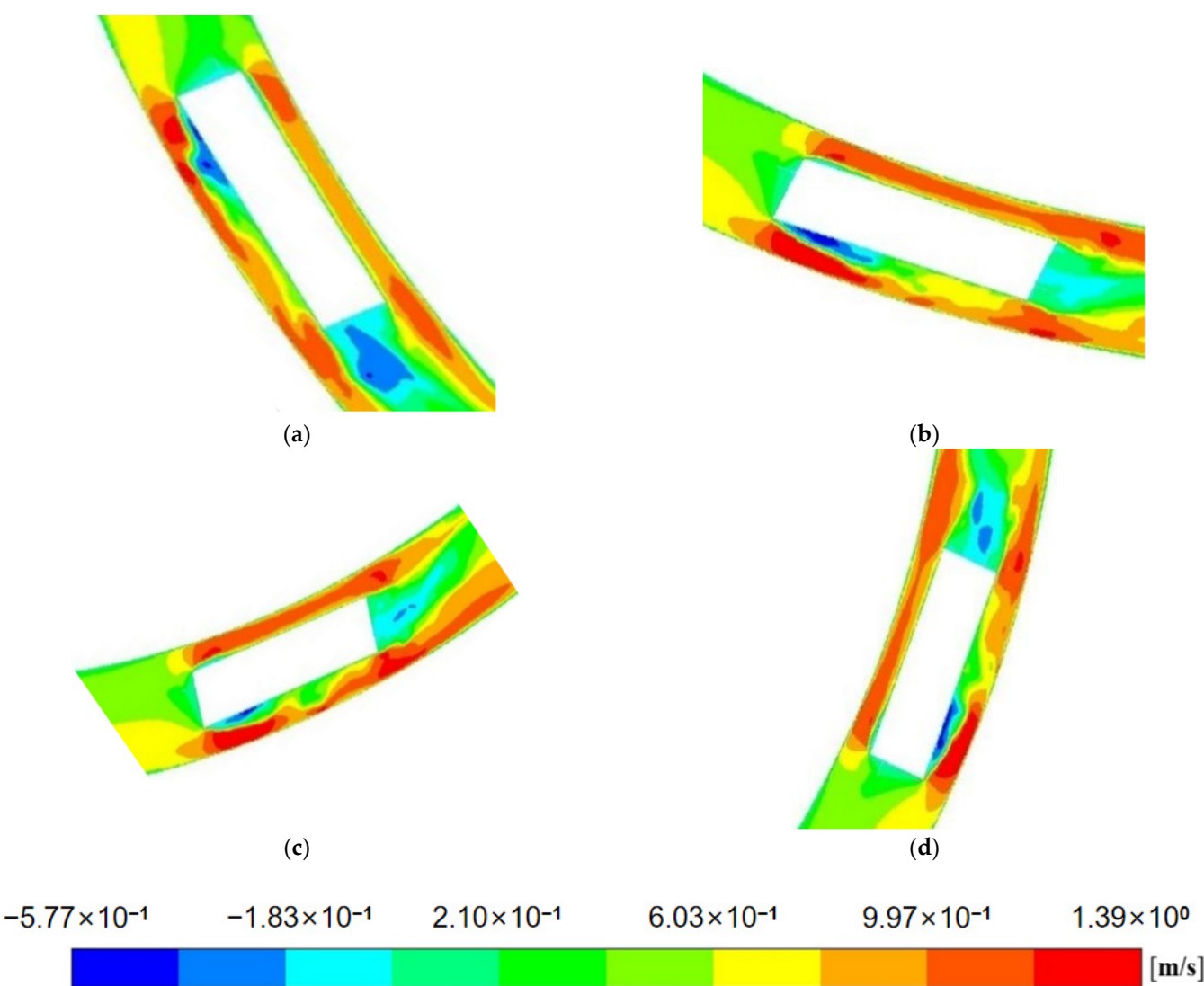

**Figure 5.** Cloud chart of the time-averaged distribution of overall capsule velocity. (**a**) Position 1; (**b**) position 2; (**c**) position 3; (**d**) position 4.

From Figure 5, it can be seen that:

The ring gap flow field distribution is more or less the same for all the different capsule starting positions on the bend. The annular gap flow field does not change significantly depending on the position of the capsule at the bend. Only the annular gap flow field distribution at position 1 is relatively chaotic compared to the other three positions. This is because the capsule at position 1 is close to the inlet section of the bend, and the bend flow has not been fully developed. The annular gap flow field distribution patterns at positions 2, 3, and 4 are similar because the bend flow at positions 2, 3, and 4 is close to the interior of the bend, and the bend flow has been fully developed. When the water reaches the ring gap area of the capsule, the flow field dramatically changes, and flow velocity suddenly increases, with greater changes on the outside of the capsule compared with the inside. This is because, as the water flows from the full pipe into the annular slit, the water crossing section suddenly becomes smaller in the process. According to the continuity equation, given a constant flow rate in a pipe, the flow rate increases when the water cross-section decreases; the flow rate will, therefore, suddenly increase when water enters the annular

gap. Due to the sudden reduction in the water cross-section, the water is then squeezed into the annular gap, and the flow line is squeezed and bent, resulting in a relatively disorderly flow field distribution. A flow of water also occurs before entering the annular gap due to the existence of the capsule support body and bypass flow. The flow of water around the support body is affected by boundary layer separation and leads to the generation of a vortex, making the flow field distribution more turbulent. In addition, because the inner side of the bend is smaller than the outer side of the bend, the angle between the flow direction and the axis of the annular gap is also smaller, so that water flow on the convex side of the bend enters the annular gap with a smaller degree of bend in the flow line. The concave side of the bend has a large bend in the flow line, and a large vortex is generated in the annular slit. The outer side of the bend near the capsule wall location can therefore be considered a low-flow-velocity zone.

### 3.2. Axial Flow Velocity Characteristics of Annular Gap Flow

In order to analyze the characteristics of the axial flow velocity at the same measurement point of the annular slit flow for different starting positions of the 1400 g loaded capsule in the bend, we set four typical measurement points [($\alpha$, $\beta$, $\gamma$), the meaning of $\alpha$ was the vertical distance from the measurement point to the horizontal plane over the capsule axis, $\beta$ was the horizontal distance from the measurement point to the vertical plane over the capsule axis, and $\gamma$ was the distance from the measurement point to the section of the inflow of the capsule slit], (0, −035, 0), (0, −040, 0), (0, 045, 0), and (0, 040, 0), in the region of the annular slit. The (0, −035, 0) and (0, −040, 0) measurement points were located on the concave side of the bend, while the (0, 045, 0) and (0, 040, 0) measurement points were located on the convex side. The 0 mm point on the horizontal axis indicated the capsule slit inflow port section, and the 150 mm point indicated the capsule slit flow outlet section. The flow velocity variation curves for each measurement point are shown in Figure 6.

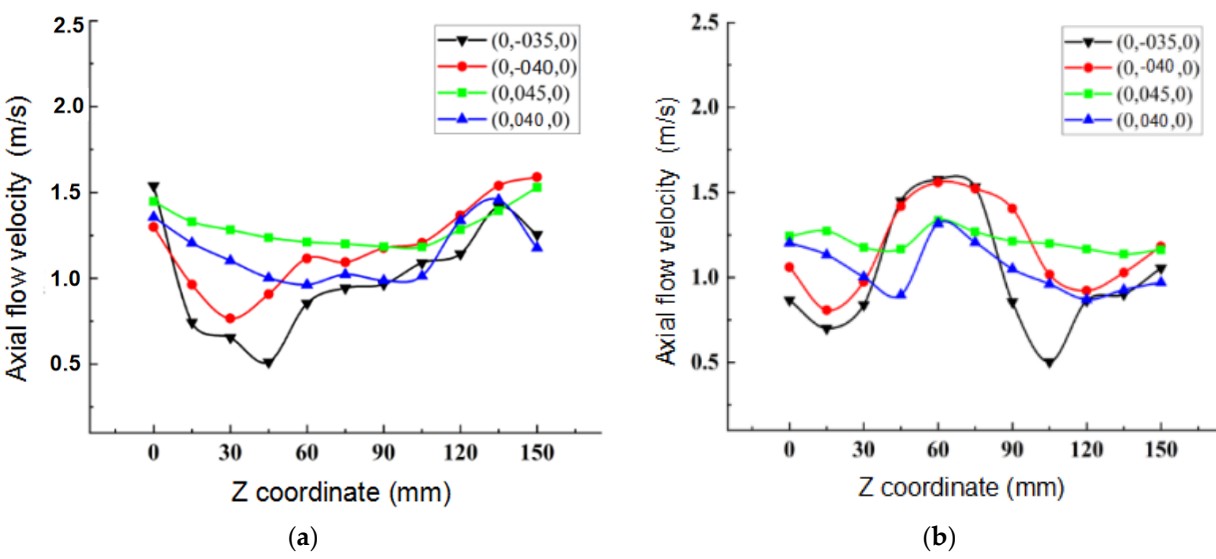

(**a**)                                                      (**b**)

**Figure 6.** *Cont.*

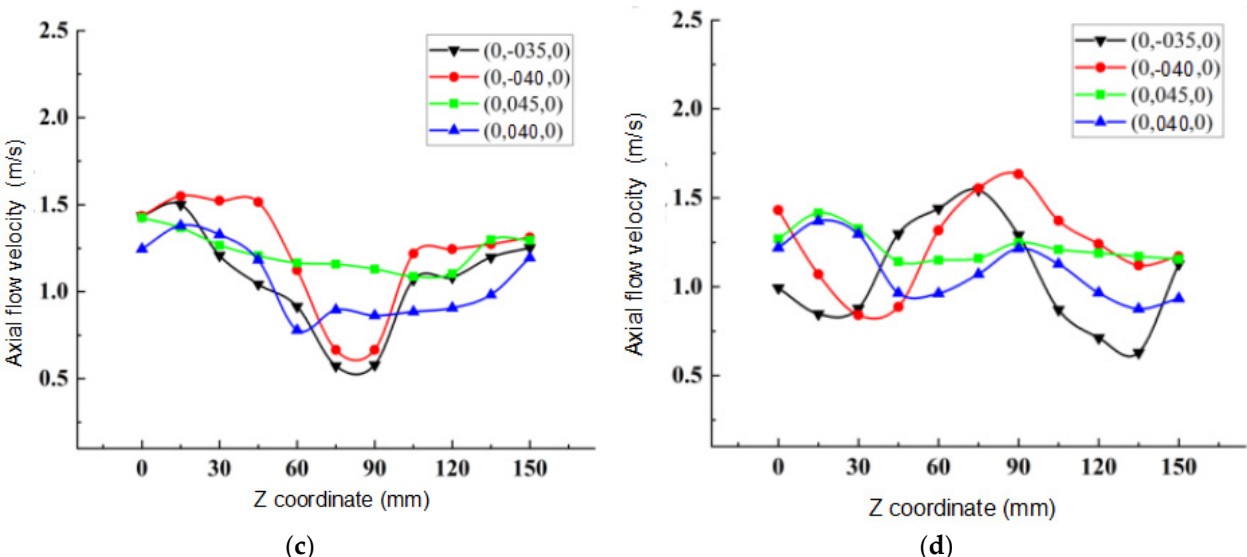

**Figure 6.** Variation curve of annular slit flow axial velocity at the same measuring point along the flow. (**a**) Position 1; (**b**) position 2; (**c**) position 3; (**d**) position 4.

From Figure 6, it can be seen that:

(1) The measurement points on the concave side of the bend and the convex side of the bend reveal different axial flow velocity changes along the course, and each exhibits its own fluctuation pattern. The measurement points on the convex side of the bend exhibit a flatter curve along the range, with a lower variation in flow velocity. The axial flow velocity on the convex side fluctuates around 1.25 m/s. The lowest value, 0.76 m/s, occurs at the 60 mm z-coordinate of the position 3 measurement point (0, 040, 0). The maximum value of 1.49 m/s occurs at the 150 mm z-coordinate of the position 1 measurement point (0, 45, 0). The maximum difference value is 0.73 m/s. On the concave side of the bend, the measurement points exhibit a more fluctuating curve along the range, and flow velocity also varies more. The minimum value of the concave-side axial flow velocity, 0.49 m/s, occurs at the 105 mm z-coordinate of the position 2 measurement point (0, −035, 0). The maximum value of 1.63 m/s occurs at the 90 mm z-coordinate of the position 4 measurement point (0, −040, 0). The maximum difference value is 1.14 m/s. The bend flow line bending degree is small on the convex side but larger on the concave side, and produces a larger vortex in the annular gap. The axial velocity change is small on the convex side of the bend water flow; along the concave side, the axial velocity fluctuates more notably.

(2) When the capsule starts at different positions on the bend, the position of the measurement point on the concave side of the bend, where the long-range axial flow velocity exhibits the greatest fluctuations, also changes. This is because, at different positions on the bend, the combined effects of the capsule and the bend on water flow are not the same. The axial flow is not the same at the different positions on the bend, and the water flow velocity reduction at the four positions of the bend on the convex side does not quite match that on the concave side.

(3) When the capsule starts at different positions of the horizontal bend, the axial flow velocity is positive and fluctuates between 0.49 m/s and 1.63 m/s, indicating that the axial flow velocity is along the direction of flow.

## 3.3. Circumferential Flow Velocity Characteristics of the Annular Gap Flow

Still using a 1400 g loaded capsule, which was set in motion at different positions of the horizontal bend, we took measurements from four typical points in the area of the annular gap for analysis purposes. A counterclockwise flow direction was considered

positive; a clockwise flow direction was considered negative. The flow velocity variation curve along the course circumference at each measurement point is shown in Figure 7.

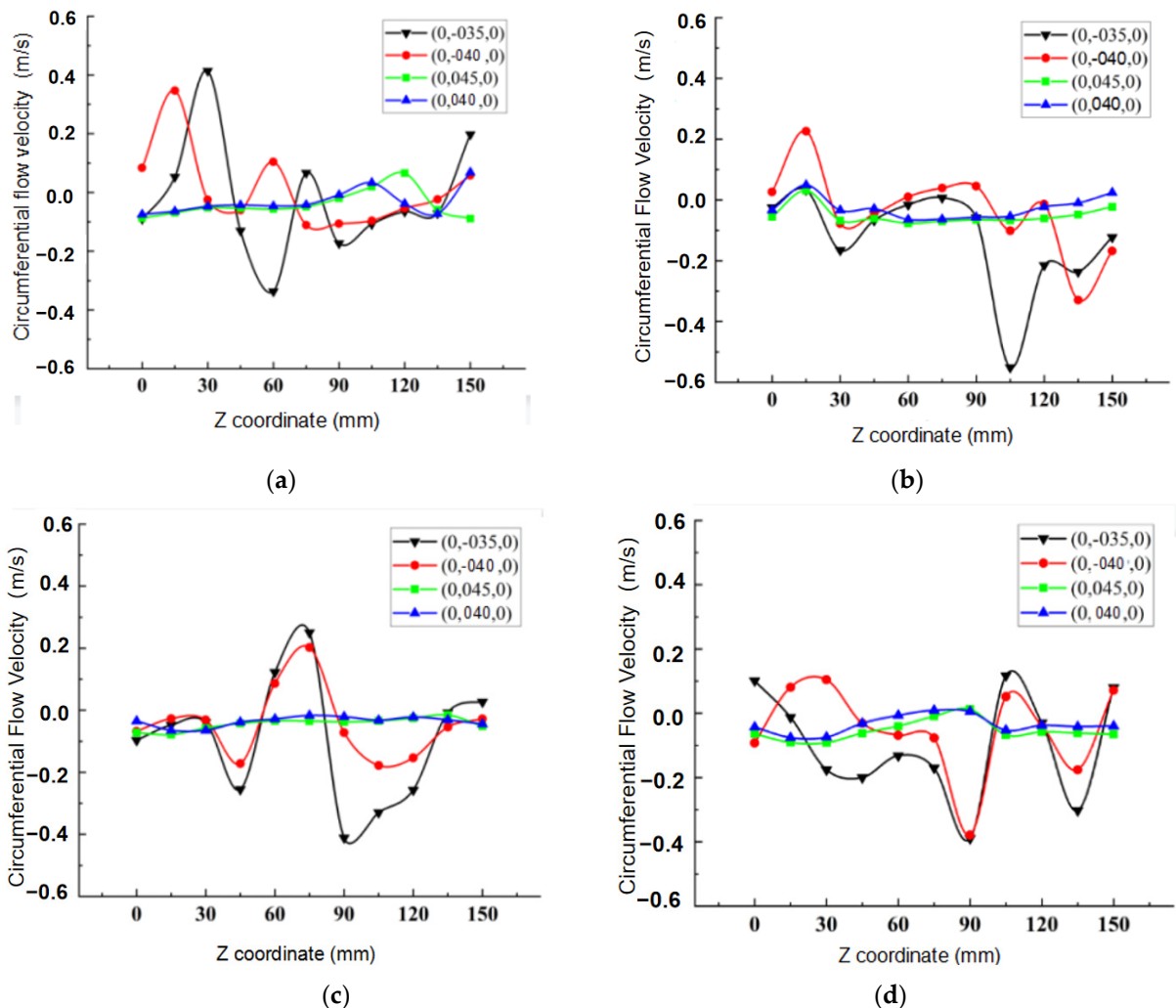

**Figure 7.** Variation curve of circumferential flow velocity at horizontal measuring points of annular slit flow. (**a**) Position 1; (**b**) position 2; (**c**) position 3; (**d**) position 4.

From Figure 7, it can be seen that:

(1) When the capsule starts at different positions of the bend, the curve of circumferential-flow velocity change along the horizontal measurement point on the convex side of the bend tends to be flat, and the flow velocity change is small. The convex-side circumferential flow velocity fluctuates around −0.05 m/s, and the difference between the maximum and minimum velocity does not exceed 0.2 m/s. This is because of the low level of water flow from the convex side of the bend into the annular gap flow line bend, which means that the change in circumferential flow velocity is not large.

(2) When the capsule starts at different positions of the bend, there are large fluctuations in the flow velocity change curve along the circumferential course at the horizontal measurement point on the concave side of the bend, and flow velocity changes are also large. The maximum value of concave-side circumferential flow velocity, 0.4 m/s, appears at the 30 mm z-coordinate of the position 1 measurement point (0, −035, 0). The minimum value of −0.58 m/s occurs at the 100 mm z-coordinate of the position 2 measurement point (0, −035, 0). The maximum difference value is 0.98 m/s. The main reason is that the water flows from the concave side of the bend into the annular gap. When the flow line bend is larger, the water in the annular gap produces a

vortex, which is positive if the direction of rotation is counterclockwise, resulting in a considerably increased flow rate. However, if the direction of rotation is clockwise, the vortex is negative, and the flow rate is greatly reduced. In a similar way to axial flow velocity, each measurement point on the concave side along the circumferential flow velocity produces large fluctuations in the location of the capsule and in the combined effects of the capsule and bend on water flow, though with notable differences at different locations on the bend.

(3) There are both positive and negative circumferential flow velocities when the capsule is activated at different positions of the horizontal bend, indicating that circumferential flow rotates in both clockwise and counterclockwise directions. The fluctuation range of circumferential flow velocity is between $(-0.58 \text{ m/s}, 0.40 \text{ m/s})$, which is about one order of magnitude smaller than that of axial flow velocity.

### 3.4. Radial Flow Velocity Characteristics of Annular Gap Flow

Again, using different starting locations of the 1400 g loaded capsule on the horizontal bend, we took measurements from four typical points in the area of the annular gap for analysis purposes. We specified a radial flow velocity along the radius of the pipe pointing to the center of the circle as negative; a velocity away from the center of the circle was specified as positive. The radial flow velocity variation curve along the range of each measurement point is shown in Figure 8.

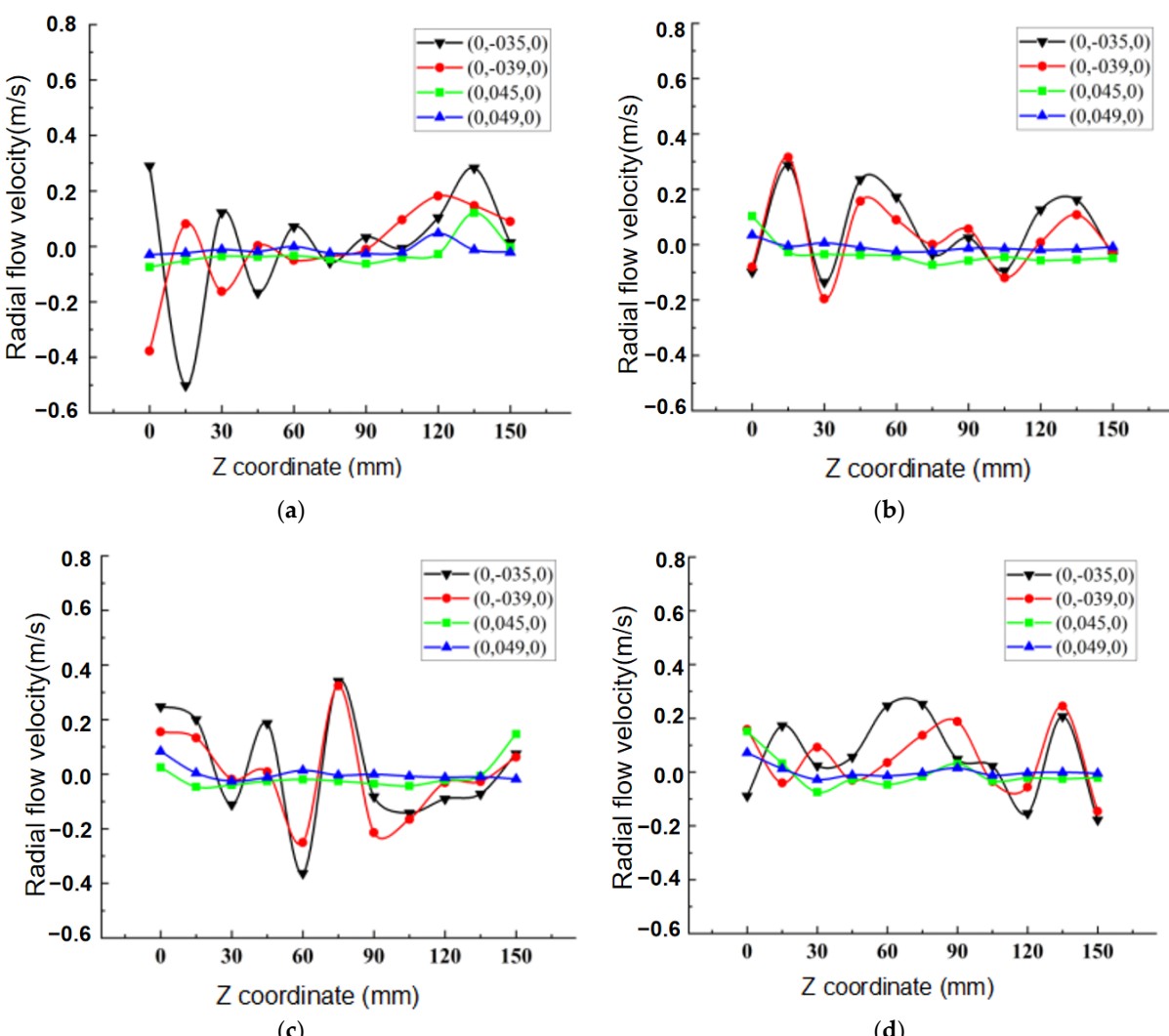

**Figure 8.** Variation curve of radial velocity of annular slit flow at the same measuring point along the flow. (**a**) Position 1; (**b**) position 2; (**c**) position 3; (**d**) position 4.

From Figure 8, it can be seen that:

(1) When the capsule starts in different positions of the bend, as with the axial and circumferential flow rates described above, the curve of radial flow rate change along the same measuring point on the convex side of the bend tends to be flat, and the flow rate change is small. The radial flow velocity on the convex side floats around 0 m/s, and the difference between the maximum value and the minimum value does not exceed 0.2 m/s. The main reason is that the water flow from the convex side of the bend into the annular gap flow line bend is small, so the radial flow rate does not change greatly.

(2) When the capsule starts at different positions of the bend, the radial flow velocity change curve along the same measurement point on the concave side of the bend exhibits large fluctuations and large flow velocity changes. The maximum value of the radial flow velocity on the concave side, 0.37 m/s, occurs at the 75 mm z-coordinate of the position 3 measurement point $(0, -035, 0)$. The minimum value of the radial flow velocity on the concave side, $-0.50$ m/s, occurs at the 15 mm z-coordinate of the position 1 measurement point $(0, -035, 0)$. The maximum difference value is 0.87 m/s. The main reason is that the flow direction of the water entering the annular gap from the concave side of the bend is more angular to the wall of the capsule, so the flow line bends more, which in turn leads to large fluctuations in flow velocity. In a similar way to the axial and circumferential flow velocities, the location of large fluctuations in radial flow velocity along the concave side of the bend at each location varies from bend to bend due to the combined effect of the capsule and the bend on the flow.

(3) As with the circumferential flow velocity, the radial flow velocity also exhibits negative values, indicating that radial flow velocity exists both in the direction pointing to the center of the circle and in the direction away from the center of the circle. The floating range is $(-0.50$ m/s,$0.37$ m/s$)$, and its order of magnitude is similar to that of the circumferential flow velocity. All of these values are lower than those for axial flow velocity by about one order of magnitude. This indicates that the flow velocity is mostly dominated by the axial flow velocity when the capsule is started at different positions in the bend.

## 4. Conclusions

(1) When the capsule is started at different positions in the bend, the difference in the flow field flow velocity distribution is small, except for at position 1, which is more turbulent because the bend current is not fully developed. When the water reaches the ring gap area of the capsule, the flow field dramatically changes, and the flow velocity suddenly increases more than in any other area.

(2) When the capsule is started on a horizontal bend, the distribution of its annular gap flow velocity is different on the concave and convex sides of the pipe due to the dual action of the bend wall, as well as the bend water flow, with each side obeying its own distribution law. On the convex side, the water flow is lower due to the flow line bend, the flow line is more gentle, and the flow velocity change is relatively small. On the concave side, the flow line bend degree is larger, a greater vortex is produced in the ring gap, the flow velocity change is more dramatic, and flow velocity distribution fluctuations are relatively large.

(3) When the capsule starts at different positions of the horizontal bend, the combined effect of the capsule and the bend on the water flow does not remain the same. The flow velocity distribution of the capsule slit flow on the concave side of the pipe varies depending on the location of the capsule. The location where the flow velocity on the concave side undergoes the greatest fluctuation also varies, depending on the location of the capsule in the pipe, although the flow velocity field distributions of the concave and convex surfaces of the pipe vary greatly at different locations of the bend. There is little variability in the distribution of the annular flow field formed by the capsule at different locations of the bend. This indicates that although the dual action of the

bend water flow as well as the bend wall will have a large effect on the flow field at different faces of the pipe, it will not change the overall distribution of the flow velocity of the capsule in the slit flow field in the bend.

(4) When the capsule is activated at different positions of the horizontal bend, the axial flow velocity is about one order of magnitude larger than the circumferential and radial flow velocities. This indicates that gap flow velocity remains dominated by the axial flow velocity for all capsule starting positions on the horizontal bend.

**Author Contributions:** Data curation, C.W.; formal analysis, C.W.; funding acquisition, C.W.; investigation, C.W.; resources, X.S.; supervision, X.S.; writing—original draft, C.W.; writing—review and editing, C.W. All authors have read and agreed to the published version of the manuscript.

**Funding:** This research was funded by the National Natural Science Foundation of China (51179116) and the Natural Science Foundation of Shanxi Province (202303021211141).

**Institutional Review Board Statement:** Not applicable.

**Informed Consent Statement:** Not applicable.

**Data Availability Statement:** This research was supported by the Collaborative Innovation Center of New Technology of Water-Saving and Secure and Efficient Operation of Long-Distance Water Transfer Project at the Taiyuan University of Technology.

**Acknowledgments:** This research was supported by the Secure and Efficient Operation of Long-Distance Water Transfer Project at the Taiyuan University of Technology.

**Conflicts of Interest:** The authors declare no potential conflict of interest with respect to the research, authorship, and/or publication of this article.

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
