# Peer review of "Experimental Analysis of the Annular Velocity of a Capsule When Starting at Different Positions of a Horizontal Bend Pipe"

_water, doi:10.3390/w15010193_

Round 1
Reviewer 1 Report
The paper presented tries to describe the annular velocity of a capsule when starting at different positions of a horizontal bend pipe. My observations on the submission are as follows:
1- Before it can be considered, the paper needs considerable revision in its English language. Extremely difficult to read, to the point where some of the ideas expressed cannot be comprehended.
2- The introduction must be updated so that it conveys not only a list of previously relevant works but also the significant findings obtained in those works and the areas that require additional research.
3- Please highlight the contrasts between your manuscript and prior works in the literature review. Please specify the knowledge gap and provide evidence that the topic is indeed unexplored. In comparison to current investigations, please list any new approaches or methodologies.
4- The manuscript appears to be about the initial conditions of a capsule's transport within the flow within a bend pipe. How plausible is this situation? Why would a capsule transport design begin with a curved segment as opposed to a normal straight section? If bends are required, they might be positioned downstream, where the capsule's motion has stabilized.
5- Practical implications should be included in the conclusion. The velocity field was evaluated, but what are the practical implications? Does something have to be changed? If so, what steps should be taken? Is it right or incorrect? How does an observation influence actual behaviour?
6-
Reviewer 2 Report
Comments to the Author
1. in the abstract , The main contribution of the paper needs to be distilled so that the reader can understand the highlights of the paper.
2. What is the innovation of this paper? How to validate your innovation?
3. In the evaluation results, how much improved the annular flow field around a capsule? Please explain in detail.
4. The authors analyzed the overall distribution of the annular gap flow velocity when the capsule started at different positions of the bend (22.5°, 67.5°, 112.5°, 167.5°). why you choose these values?
5- why you used 1400g-loaded capsule?
6- Figures 5 and 6 need to be resized.
7- Conclusions should be rewritten to provide more information about the concluded points
8. Literature should be checked if there are no newer items. Especially from the last 18 months.
9. please provide more information about the practical applications of implementing the proposed method.
Reviewer 3 Report
The presentation format of this article is far from the presentation format of a standard scientific paper. The abstract of the article, the purpose of the study, the methods applied and the parameters studied and the working ranges of parameters considered should be stated.
In the last paragraph of the introduction, the aim of this study is expected to explain the importance of the method applied and the expected outputs should be stated.
The importance of the subject of the article and its contribution to the literature should be summarized
In the manuscript, it was stated that the PIV method was used in the experimental study. However, no information is provided about the use of the PIV method. For example, in the elbow section, how to measure the velocity field over the cross-sectional area of the pipe should be explained with sketches.
For the reader to understand the text part of the manuscript better and faster, it is necessary to support the given information with figures.
PIV generally measures the instantaneous velocity distribution in a given flow area. It is unclear how the raw data were processed.
What is the measuring frequency of PIV?
Why is the particle sizes quite large?
Also, the laser beam thickness is taken as five mm. What is the error rate of the third dimension of the flow when measuring velocity distributions in a two-dimensional flow field? Generally, the thickness of the laser flow beam is taken to be around 2 mm for further accurate velocity measurement.
To provide further information about the mechanisms of the flow, one needs to have further instantaneous flow data since the PIV method can also provide instantaneous flow data over the measuring flow area.
It is known that in the elbow there is usually secondary flow. But, in the manuscript, there is no information about the secondary flow.
Three is no data given about turbulent statistics.
Please prepare the schematic diagrams of instruments and test setup. And also prepare the view of the test section.
Ä°t is better to use to convert measured parameters into dimensionless form.
Round 2
Reviewer 1 Report
All comments had been addressed. it can be recommended for publication.
Reviewer 3 Report
The necessary revision has been conducted in line with the referees' suggestions. Now, the manuscript deserves to be published in your journal.